

# Cell proliferation controls body size growth, tentacle morphogenesis, and regeneration in hydrozoan jellyfish *Cladonema pacificum*

Sosuke Fujita[1], Erina Kuranaga[1] and Yu-ichiro Nakajima[1,2]

[1] Graduate School of Life Sciences, Tohoku University, Sendai, Japan
[2] Frontier Research Institute for Interdisciplinary Sciences, Tohoku University, Sendai, Japan

## ABSTRACT

Jellyfish have existed on the earth for around 600 million years and have evolved in response to environmental changes. Hydrozoan jellyfish, members of phylum Cnidaria, exist in multiple life stages, including planula larvae, vegetatively-propagating polyps, and sexually-reproducing medusae. Although free-swimming medusae display complex morphology and exhibit increase in body size and regenerative ability, their underlying cellular mechanisms are poorly understood. Here, we investigate the roles of cell proliferation in body-size growth, appendage morphogenesis, and regeneration using *Cladonema pacificum* as a hydrozoan jellyfish model. By examining the distribution of S phase cells and mitotic cells, we revealed spatially distinct proliferating cell populations in medusae, uniform cell proliferation in the umbrella, and clustered cell proliferation in tentacles. Blocking cell proliferation by hydroxyurea caused inhibition of body size growth and defects in tentacle branching, nematocyte differentiation, and regeneration. Local cell proliferation in tentacle bulbs is observed in medusae of two other hydrozoan species, *Cytaeis uchidae* and *Rathkea octopunctata*, indicating that it may be a conserved feature among hydrozoan jellyfish. Altogether, our results suggest that hydrozoan medusae possess actively proliferating cells and provide experimental evidence regarding the role of cell proliferation in body-size control, tentacle morphogenesis, and regeneration.

## INTRODUCTION

Cell proliferation lies at the core of controlling cell number in Metazoa and thus contributes to the growth and the maintenance of animal body and organs (*Leevers & McNeill, 2005*; *Penzo-Méndez & Stanger, 2015*). During development, cell proliferation plays a critical role in body-size increase by adding cells into tissue layers, and it further generates cellular resources for different cell types by multiplying progenitors (*Gillies & Cabernard, 2011*; *Hardwick et al., 2015*). Later in adults, proliferating cells are required for physiological cell turnover and for the replacement of damaged cells after tissue injury

Corresponding author
Yu-ichiro Nakajima,
yuichiro.nakajima.d2@tohoku.ac.jp

(*King & Newmark, 2012*; *Pellettieri & Sanchez Alvarado, 2007*). These roles of cell proliferation in multicellularity must be conserved throughout evolution: indeed, sponges, one of the earliest metazoan organisms, have acquired mechanisms to allow cell turnover by controlling proliferative capacities (*Alexander et al., 2014*; *Kahn & Leys, 2016*).

As the sister group of bilaterians and early-branching metazoans, cnidarians have been studied as a model to understand evolutionary development (*Genikhovich & Technau, 2017*). Cnidarians are diploblastic and radially symmetric animals that include diverse species such as corals, sea anemones, hydroids, and jellyfish (*Technau & Steele, 2011*). During the embryonic development of the sea anemone *Nematostella vectensis*, cell proliferation is coordinated with epithelial organization and is involved in tentacle development (*Fritz et al., 2013*; *Ragkousi et al., 2017*). Cnidarians are also known for their regenerative abilities: for instance, *Hydra* polyps have been used for a century to investigate mechanisms of metazoan regeneration (*Fujisawa, 2003*; *Galliot & Schmid, 2002*). The basal head regeneration of *Hydra* relies on cell proliferation triggered by dying cells (*Chera et al., 2009b*; *Galliot & Chera, 2010*). *Hydractinia* polyps regenerate through cell proliferation and the migration of stem-like cells (*Bradshaw, Thompson & Frank, 2015*; *Gahan et al., 2016*). Although much has been learned about mechanisms controlling embryogenesis and growth during regeneration, it is unclear how cnidarians integrate cell proliferation to control their body size and maintain tissue homeostasis under normal physiological conditions.

Among cnidarians, hydrozoan jellyfish have a complex life cycle including planula larvae, sessile polyps, and free-swimming medusae. While polyps undergo asexual reproduction to grow vegetatively, medusae generate gametes to perform sexual reproduction. Despite the limited life span compared to the long-lived or possibly immortal polyps, the size of medusae increases dramatically (*Hansson, 1997*; *Miyake, Iwao & Kakinuma, 1997*). Furthermore, medusae maintain their regenerative capacity for missing body parts by integrating dedifferentiation and transdifferentiation (*Schmid & Alder, 1984*; *Schmid et al., 1988*; *Schmid, Wydler & Alder, 1982*). Recent studies using the hydrozoan jellyfish *Clytia hemisphaerica* have provided mechanistic insights into embryogenesis, nematogenesis, and egg maturation (*Denker et al., 2008*; *Momose, Derelle & Houliston, 2008*; *Quiroga Artigas et al., 2018*). However, little is known about the mechanism that controls body size growth in medusae. It is also unclear whether cell proliferation is required for tentacle morphogenesis and regeneration of hydrozoan jellyfish.

The hydrozoan jellyfish *Cladonema* is an emerging model, with easy lab maintenance and a high spawning rate, that is suitable for studying diverse aspects of biology including development, regeneration, and physiology (*Fujiki et al., 2019*; *Graziussi et al., 2012*; *Suga et al., 2010*; *Takeda et al., 2018*; *Weber, 1981*). *Cladonema* is characterized by small-sized medusae with branched tentacles. Using specialized adhesive tentacles, *Cladonema* can adhere to different substrata, such as seaweed, in the field. The species *Cladonema pacificum*, originally found along coastal areas in Japan, have nine main tentacles with a stereotyped branching pattern (Figs. 1A–1C). During the *Cladonema medusa's* maturation, body size increases, and each main tentacle grows and exhibits

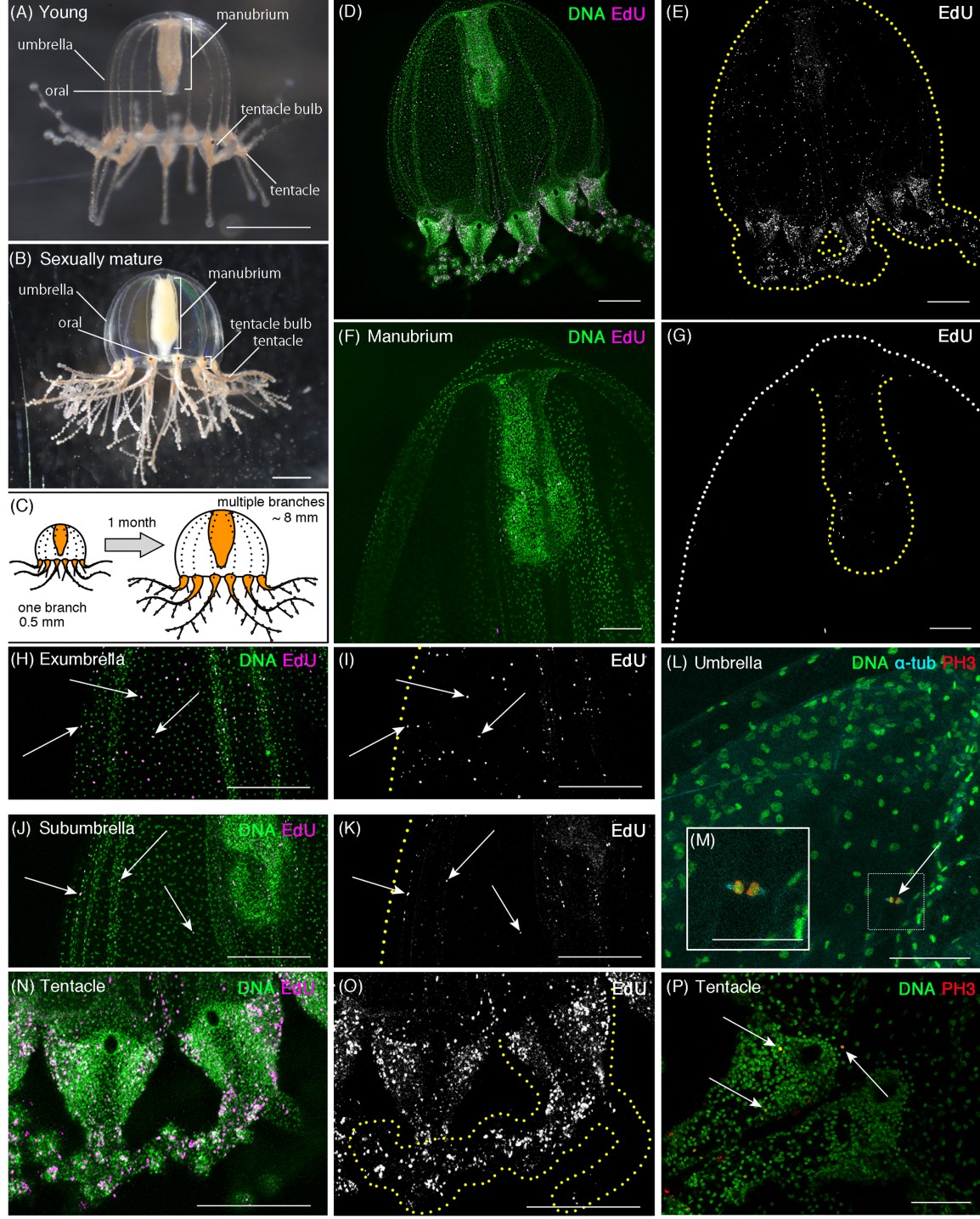

**Figure 1 Cell proliferation patterns in young *Cladonema* medusa.** (A) Young medusa of *Cladonema pacificum*. (B) Sexually-matured medusa of *Cladonema pacificum*. (C) The scheme of *Cladonema* medusa development. (D–K, N, O) Distribution of S-phase cells in the *Cladonema pacificum* medusa (1 day old) revealed by EdU staining (20 μM, 24 h incubation). (D, E) Distribution of S-phase cells (EdU+) in a medusa whole body. (F, G) Distribution of S-phase cells (EdU+) in a medusa manubrium. (H, I) Distribution of S-phase cells (EdU+) in a medusa exumbrella. (J, K) Distribution of S-phase cells (EdU+) in a medusa subumbrella. (L, M) Mitotic cells detected by anti-PH3 in a medusa umbrella (8 day old). (N, O) Distribution of S-phase cells (EdU+) in medusa tentacles. (P) Mitotic cells (PH3+) in medusa tentacle bulbs (1 day old). Arrows indicate EdU-positive (H–K) and PH3-positive (L, P) cells, respectively. Scale bars: (A, B) one mm, (D, E, H–K, N, O) 200 μm, (F, G) 100 μm, (L, M, P) 50 μm.

branching morphology (*Fujiki et al., 2019*), providing an ideal system to dissect the cellular mechanisms associated with jellyfish growth and morphogenesis.

In this study, we investigate the role of cell proliferation in medusa growth and morphogenesis, using *Cladonema pacificum* as a model of hydrozoan jellyfish. We show that cell proliferation occurs evenly across the medusa body, including the umbrella and manubrium, with the exception of the tentacles, where cell proliferation is spatially clustered. Blocking cell-cycle progression with a pharmacological assay inhibits the increase of body size, tentacle branching, and nematocyte differentiation, which suggests that cell proliferation is necessary for growth and tentacle morphogenesis. We further show that cell proliferation is required for tentacle regeneration in *Cladonema* medusae. Our findings reveal cell proliferation's critical roles in the development and maintenance of the *Cladonema* body and appendages and provide a basis for understanding growth-control mechanisms in hydrozoan jellyfish.

# MATERIALS AND METHODS

## Animal cultures

We used *Cladonema pacificum* (strains 6W and UN2) (Figs. 1–5; Figs. S1–S3), *Cytaeis uchidae* (strain ♀17) (Fig. 6) and *Rathkea octopunctata* (strain MF-1) (Fig. 6) medusae for this research. The medusae were cultured in plastic cups (V-type container, V-7 and V-8, AS ONE, Osaka, Japan) at 20 °C (*Cladonema* and *Cytaeis*) or 4 °C (*Rathkea*), and their polyps were maintained in the cups (V-7) at 20 or 4 °C in darkness. Vietnamese brine shrimp (A&A Marine LLC, Elk Rapids, MI, USA) were fed to medusae and polyps every other day ad libitum, with water renewed immediately after feeding. Artificial sea water (ASW) was prepared by SEA LIFE (Marin Tech, Tokyo, Japan). Pictures of medusae were taken through a LEICA S8APO microscope with a Nikon digital camera (D5600).

## Immunofluorescence

The medusae were anesthetized with 7% $MgCl_2$ in ASW for 10 min and fixed 4% paraformaldehyde (PFA) in ASW for 1 h. After fixation, the samples were rinsed in 1× PBS and washed three times (10 min each) in PBS containing 0.1% Triton X-100 (0.1% PBT). The samples were incubated in primary antibodies in 0.1% PBT overnight at 4 °C. The antibodies used were rabbit anti-Phospho-Histone H3 (Ser10) (1:500; 06–570, Upstate, Lake Placid, NY, USA) and mouse anti-α-Tubulin (1:500; T6199, Sigma-Aldrich, St Louis, MO, USA). After the primary antibody incubation, the samples were washed three times (10 min each) in 0.1% PBT and incubated in secondary antibodies (1:500; ALEXA FLUOR Goat anti-mouse IgG, ALEXA FLUOR Goat anti-rabbit IgG; Thermo Fisher Scientific, Waltham, MA, USA) and Hoechst 33342 (1:250; Thermo Fisher Scientific, Waltham, MA, USA) in 0.1% PBT for 1 h in dark. After four washes (10 min each) in 0.1% PBT, the samples were mounted on slides with 70% glycerol. Confocal images were collected through Leica SP8 or SP5 confocal microscopes. Z-stack images were performed using ImageJ/Fiji software.

## EdU labeling

The medusae were incubated with 20 µM 5-ethynyl-2′-deoxyuridine (EdU; EdU kit; 1836341; Invitrogen, Carlsbad, CA, USA) in ASW for 24 h (Figs. 1–3 and 6) or 150 µM for

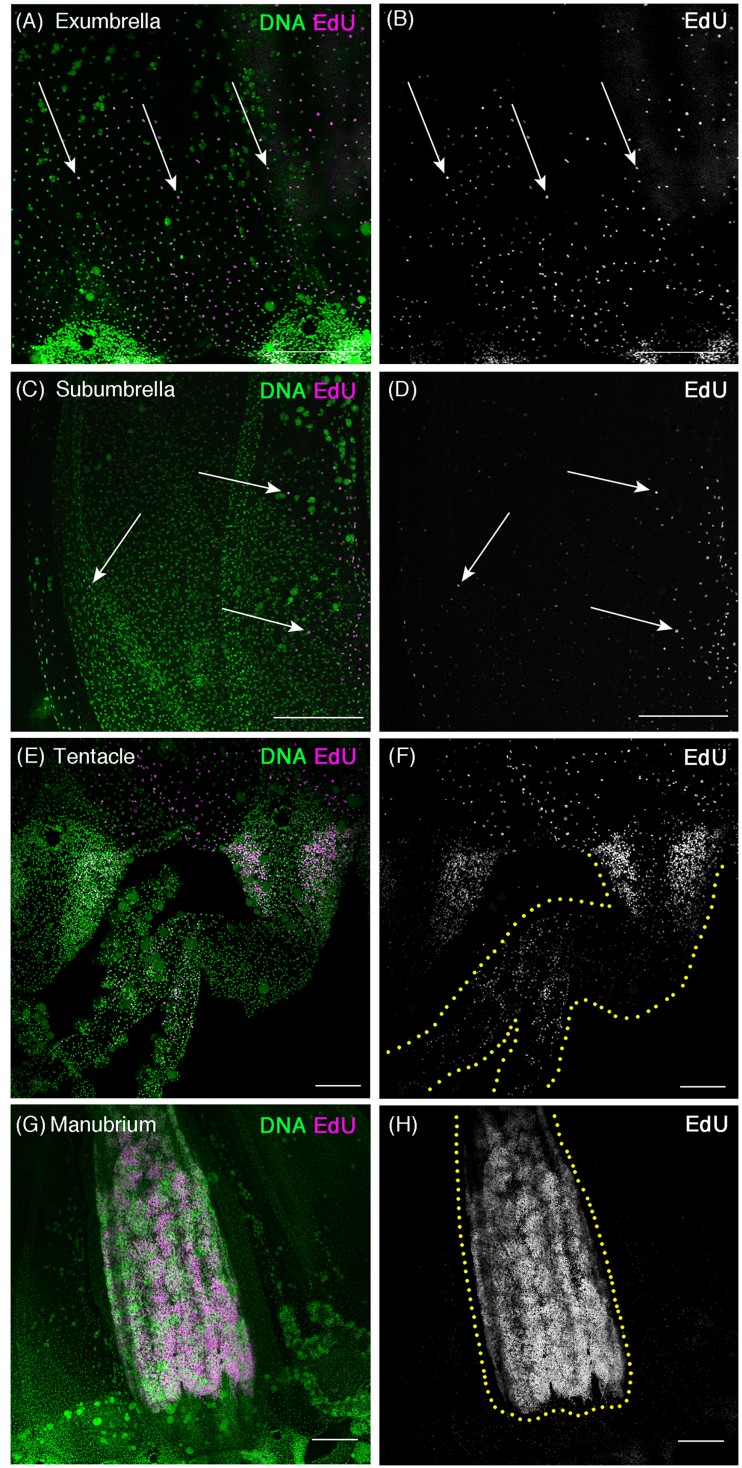

**Figure 2 Cell proliferation patterns in sexually mature *Cladonema* medusa.** (A–H) Distribution of S-phase cells in the *Cladonema pacificum* medusa (45 day old) shown with EdU staining (20 μM, 24 h incubation). (A, B) Distribution of S-phase cells (EdU+) in a medusa exumbrella. (C, D) Distribution of S-phase cells (EdU+) in a medusa subumbrella. (E, F) Distribution of S-phase cells (EdU+) in medusa tentacles. (G, H) Distribution of S-phase cells (EdU+) in a medusa manubrium. Arrows indicate EdU-positive (A–D) cells. Scale bars: (A–D, G, H) 200 μm, (E, F) 100 μm.

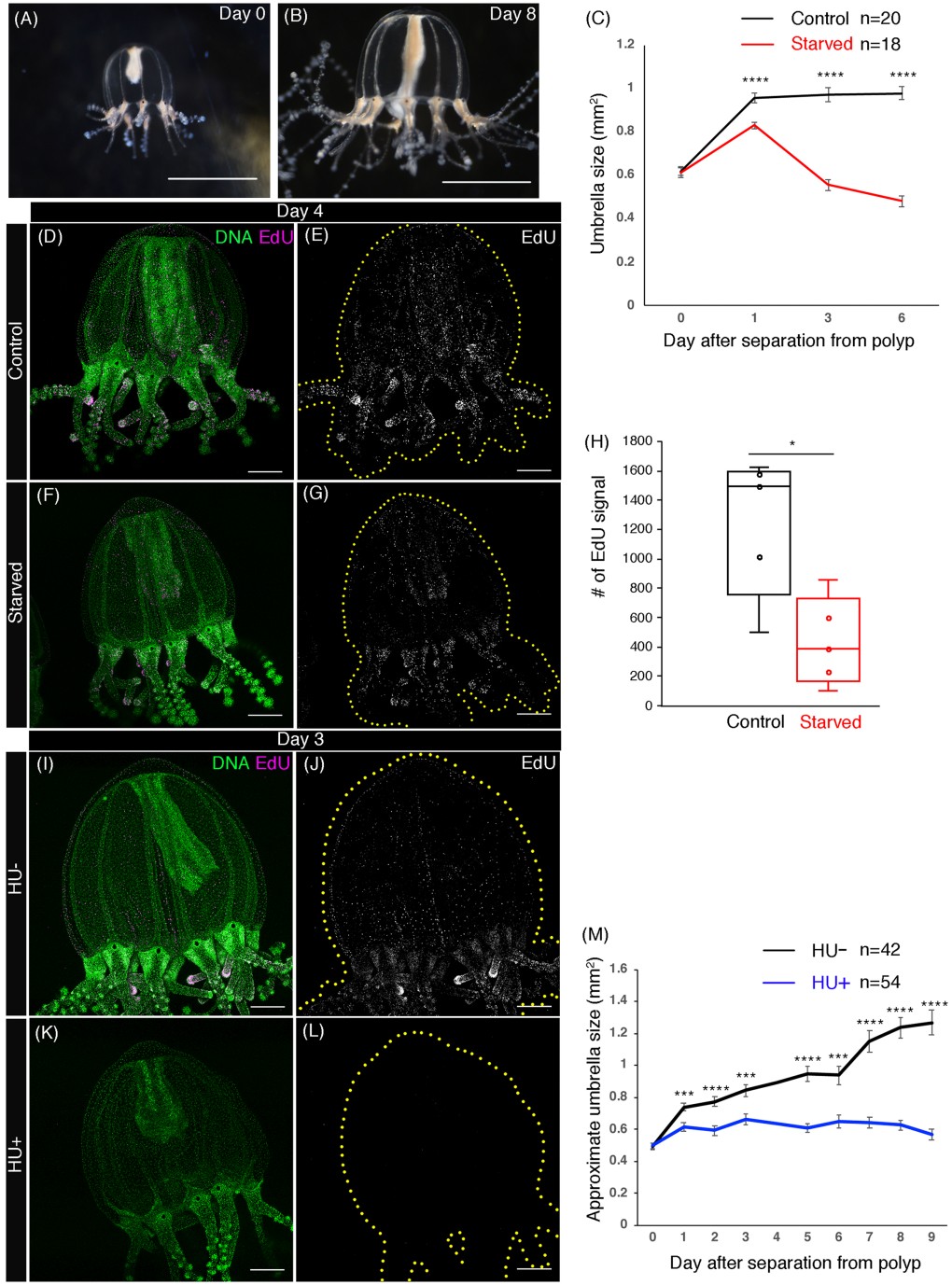

**Figure 3 Cell proliferation is necessary for body-size growth.** (A) *Cladonema pacificum* newborn medusa (0 day old). (B) *Cladonema pacificum* juvenile medusa (8 day old). (C) Quantification of umbrella size in control and starved medusae. Control medusae were fed every other day. Error bar: SD. Unpaired two-tailed *t*-test. Day 1 $t(36) = 4.545$, day 3 $t(36) = 9.888$, day 6 $t(36) = 12.56$, ****$p < 0.0001$. (D–G) Distribution of S-phase cells in control medusa and starved medusa with EdU staining (20 µM, 24 h incubation). (H) Quantification of the number of S-phase cells (EdU+) in control and starved medusae. Unpaired two-tailed *t*-test. *$p < 0.05$ ($p = 0.0127$), $t = 3.194$ d$f = 8$. (I–L) Distribution of S-phase cells in medusa of control (HU−) and hydroxyurea (HU+) treatment detected by EdU staining (20 µM, 24 h). No S-phase cells were detected in HU+ medusae. (M) Quantification of body size in control and in HU conditions. HU suppresses body-size growth. HU−: control medusae incubated in ASW, HU+: medusae incubated in HU 10 mM ASW. Both HU+ and HU− were fed every other day. Error bar: SD. Unpaired two-tailed *t*-test. Day 1 $t(93) = 3.561$, day 2 $t(90) = 4.079$, day 3 $t(81) = 3.657$, day 5 $t(85) = 6.329$, day 6 $t(52) = 4.105$, day 7 $t(79) = 7.319$, day 8 $t(71) = 9.201$, day 9 $t(59) = 8.826$, ***$p < 0.0005$, ****$p < 0.0001$. Scale bars: (A, B) one mm, (D–G and I–L) 100 µm.

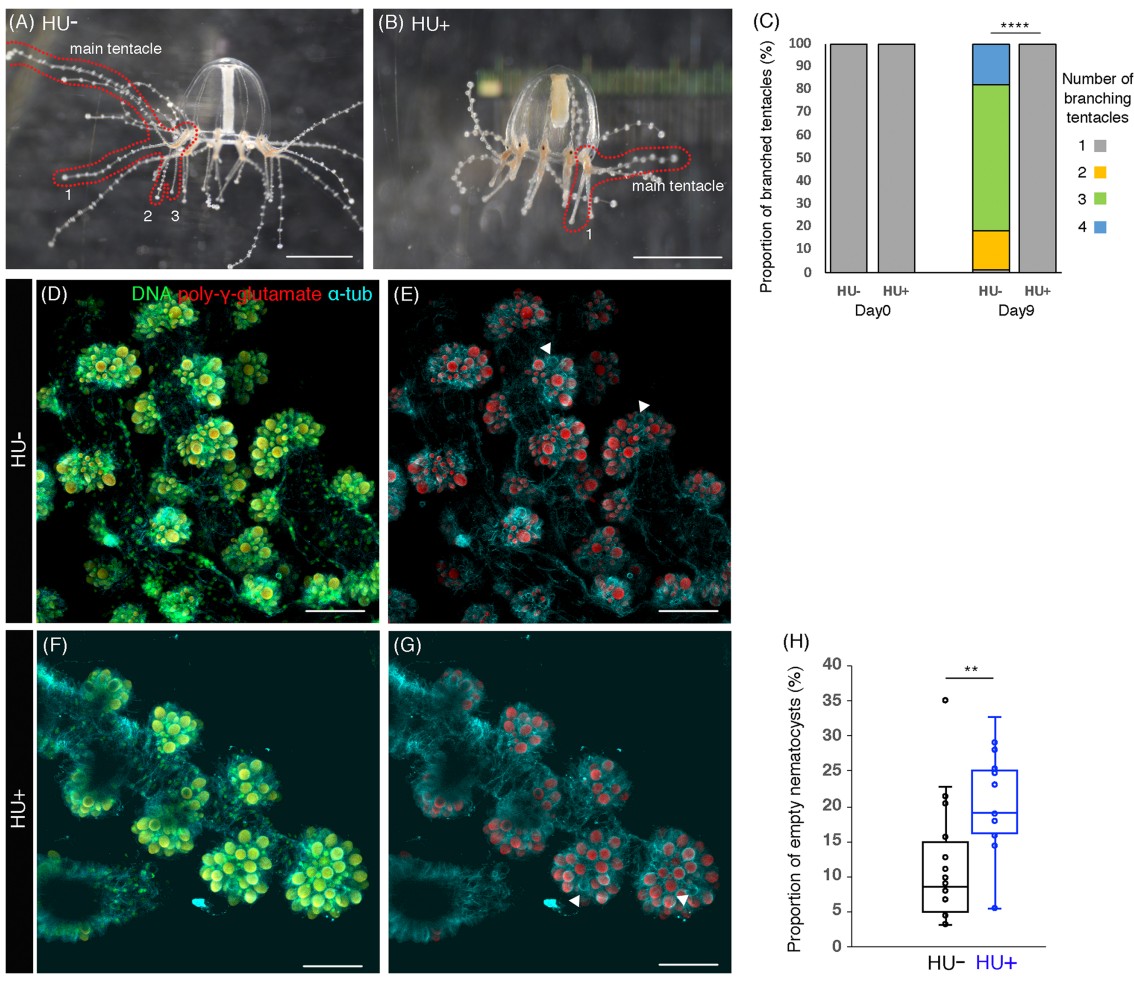

**Figure 4  Cell proliferation is necessary for tentacle morphogenesis.** (A) Control (HU−) medusa incubated in ASW for 9 days. The picture shows the representative image of medusae with three branched tentacles. (B) The medusa incubated in 10 mM HU (HU+) ASW for 9 days. The picture shows the representative image of medusae with one branched tentacle. (C) Quantification of branching numbers per tentacle at day 0 and day 9. HU+: $n = 313$, HU− condition: $n = 199$. Error bars: SD. Unpaired two-tailed $t$-test. $t(510) = 54.49$, ****$p < 0.0001$. (D–G) Nematocytes in tentacles labeled by DAPI (poly-γ-glutamate) in the 8 day old medusa incubated in ASW (HU−) or 10 mM HU ASW (HU+). Arrowheads indicate empty nematocysts. (H) The proportion of empty nematocysts in HU− and HU+ medusa. HU+: $n = 19$, HU−: $n = 18$. Unpaired two-tailed $t$-test. $t(31) = 2.869$, **$p < 0.01$ ($p = 0.0074$). Scale bar: (D–G) 50 μm.

1 h (Fig. S1). After EdU treatment, the medusae were anesthetized with 7% MgCl$_2$ in ASW for 10 min and fixed 4% PFA in ASW for 1 h. After fixation, the samples were rinsed in 1× PBS and washed three times (10 min each) in 0.1% PBT. The samples were incubated with a EdU reaction cocktail (1× reaction buffer, CuSO$_4$, Alexa Fluor azide, and 1× reaction buffer additive; all included in EdU kit; 1836341; Invitrogen, Carlsbad, CA, USA) for 30 min in the dark. After the EdU reaction, the samples were washed three times (10 min each) in 0.1% PBT and Hoechst 33342 (1:250; Thermo Fisher Scientific, Waltham, MA, USA) in 0.1% PBT for 1 h in dark. The samples were washed four times (10 min each) in 0.1% PBT and were mounted on slides with 70% glycerol.

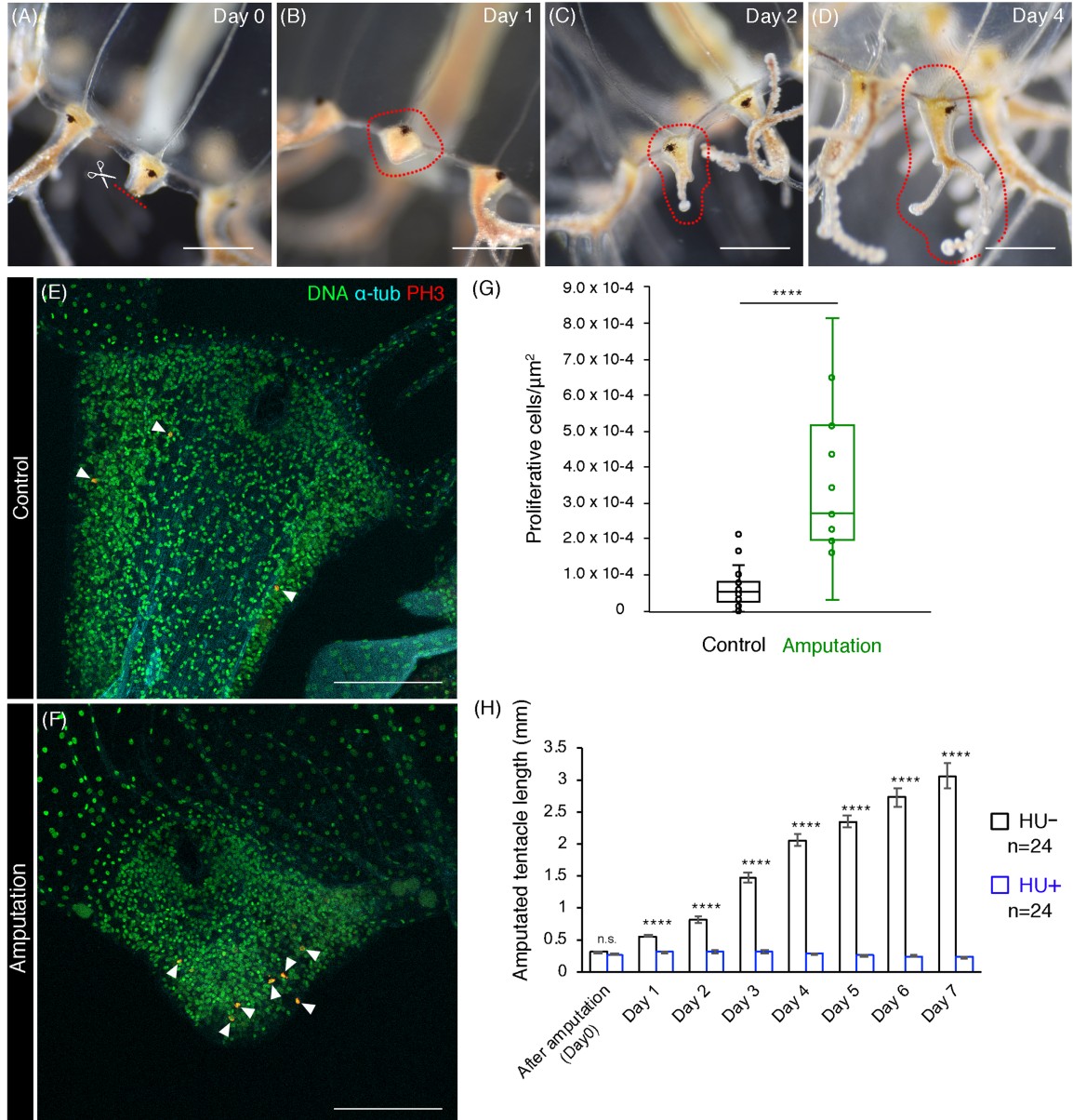

**Figure 5 Cell proliferation is necessary for tentacle regeneration.** (A–D) Tentacle regenerative processes after amputation in an adult medusa. Series of pictures show the growing tentacle over 4 days. (E, F) Mitotic cells (PH3+) in tentacle bulbs of the unremoved control and the dissected medusa. Arrowheads indicate PH3-positive cells. (G) Quantification of proliferative cells in tentacle bulbs for control and after amputation. Control: $n = 26$, Amputation: $n = 11$. Error bar: SD. Unpaired two-tailed $t$-test. $t(35) = 6.246$, ****$p < 0.0001$. (H) Quantification of tentacle length after amputation in control (HU−) and 10 mM HU treatment (HU+). Unpaired two-tailed $t$-test. Day 1 $t(46) = 9.227$, day 2 $t(46) = 10.29$, day 3 $t(46) = 14.1$, day 4 $t(46) = 20.5$, day 5 $t(46) = 22.49$, day 6 $t(45) = 17.11$, day 7 $t(45) = 15.36$, ****$p < 0.0001$. Scale bars: (A–D) one mm, (E, F) 100 μm.

## Hydroxyurea treatment

The live medusae were incubated with 10 mM hydroxyurea (HU) (085-06653; Wako, Osaka, Japan) in ASW (ASW only for control) (Figs. 3–5; Fig. S3). HU incubation was continued for a maximum of 9 days. Medusae were fed every other day, and HU solution or ASW was

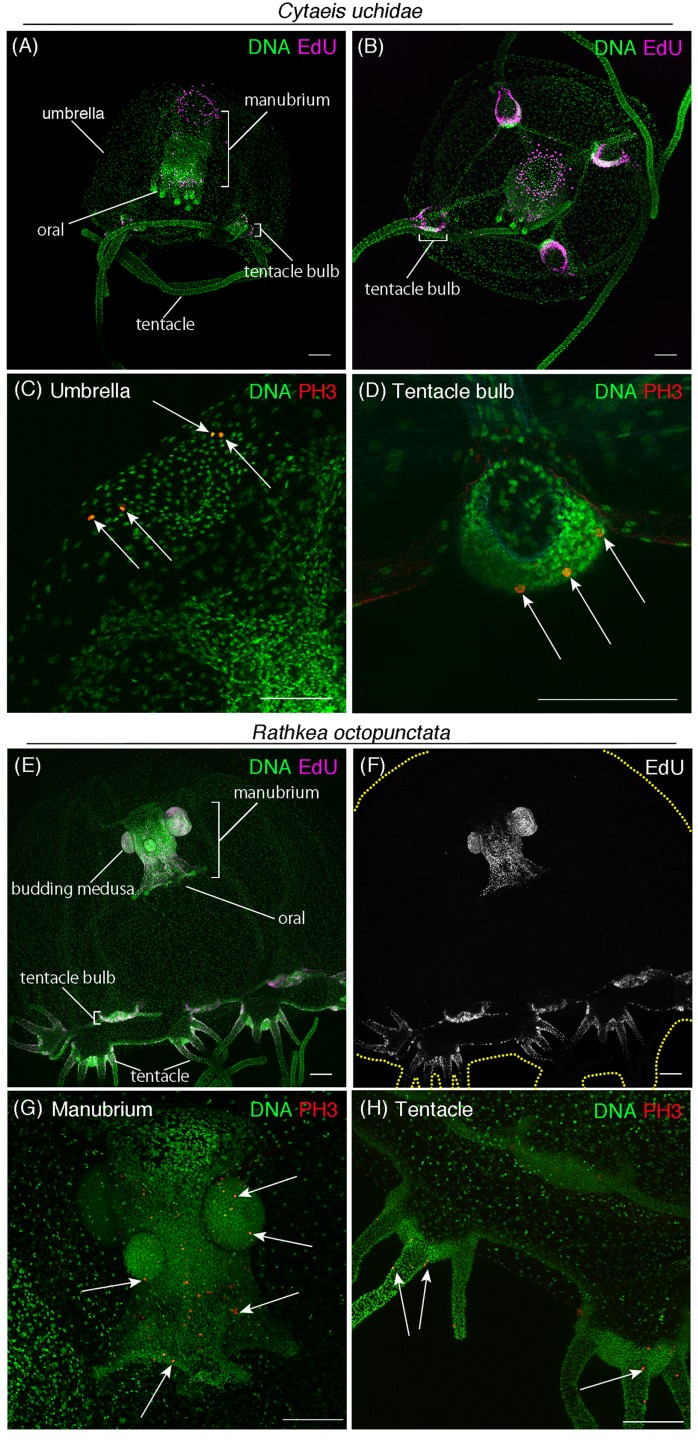

**Figure 6 Cell proliferation patterns across different hydrozoan jellyfish.** (A) Distribution of S-phase cells in the *Cytaeis uchidae* medusa (30 day old) shown with EdU staining (EdU: 20 μM, 24 h). (B) Distribution of S-phase cells (EdU+) in *Cytaeis* medusa (11 day old). (C) Mitotic cells (PH3+) in the umbrella of *Cytaeis* medusa (30 day old). (D) Mitotic cells in *Cytaeis* medusa tentacle bulbs (30 day old). (E, F) Distribution of S-phase cells (EdU+) in the *Rathkea octpunctata* juvenile medusa (EdU: 20 μM, 24 h). (G) Mitotic cells (PH3+) in a manubrium of *Rathkea* juvenile medusa. (H) Mitotic cells (PH3+) in *Rathkea* juvenile medusa tentacles. Arrows indicate PH3-positive mitotic cells. Scale bars: 100 μm.

renewed after feeding. The medusae treated with HU were able to ingest prey like controls, demonstrating that HU treatment had no effect on feeding behavior (Figs. S3A and S3B).

### Measurement of umbrella size and tentacle length

Pictures of medusae were taken with a Nikon D5600, and umbrella size was measured using polygon selections with ImageJ software (Fig. 3C). We measured the length and width of medusae under the microscope using an ocular micrometer and multiplied the length and width to generate a value for umbrella size (Fig. 3M). Tentacle length was measured daily under the microscope with an ocular micrometer (Fig. 5H).

### DAPI poly-γ-glutamate staining

This protocol was adapted from *Szczepanek, Cikala & David (2002)*: The medusae were anesthetized with 7% $MgCl_2$ in ASW for 10 min and fixed with 4% PFA in ASW for 1 h. After fixation, the samples were rinsed in 1× PBS and washed three times (10 min each) in 0.1% PBT. The samples were incubated in DAPI (1:500; Polysciences, Inc., Warrington, PA, USA) in PBT for 60 min. After the DAPI incubation, samples were washed four times (10 min each) in PBT and mounted on slides with 70% glycerol in DW. Samples were scanned with a combination of 488 nm excitation and 555 nm emission filter using either Leica SP8 or SP5 confocal microscopes. Using ImageJ, we performed Z-stacks and counted nematocysts. Empty nematocysts were counted manually.

### Dissection of tentacles for regeneration

Tentacles' basal sides were dissected with small scissors, leaving the tentacle bulbs intact. Amputated medusae were fed every other day.

### Statistical test

An unpaired two-tailed *t*-test was performed on the data shown in Figs. 3–5.

### Average nearest neighbor distance (spatial statistics)

We performed statistical analysis for the proliferating cells' distribution in umbrellas and tentacles by applying the nearest neighbor distance (NND) test to EdU positive cells (Table S1). Here, we used the images of 1-day-old medusae that had been incubated with EdU 150 μM for 1 h. This analysis was applied to the umbrella area, except for tentacle bulb and manubrium, while the same analysis was applied to the entire main tentacle. The area (S), the signal number (N), and NND in analyzed areas were obtained using ImageJ/Fiji. The average of NND (W), the expectation value of W (E[W]), and the normalized average of NND ($w$ = W/E[W]) were calculated. In this analysis, $w > 1$ means that EdU signals are distributed uniformly or randomly. In contrast, $w < 1$ means that EdU signals are distributed clustered or randomly. The spatial distribution of EdU signals were determined by Z score.

## RESULTS

### Cell proliferation patterns in the medusa *Cladonema pacificum*

To understand the spatial pattern of cell proliferation in *Cladonema* medusa, we performed EdU staining, which labels S-phase or the former S-phase cells (*Salic &*

*Mitchison, 2008*). Given that *Cladonema* medusa dramatically increases in size and exhibits tentacle branching during development (Figs. 1A–1C), distribution of proliferating cells could change throughout maturation. We thus investigated cell proliferation patterns in both young (day 1) and sexually mature (day 45) medusae.

In young medusae, EdU-positive cells were broadly detected in the whole medusa body including the umbrella, the manubrium (a supporting organ of the oral in medusae), and the tentacles regardless of the incubating time of EdU (Figs. 1D–1K and 1N–1O, EdU: 20 µM for 24 h; Fig. S1, EdU: 150 µM for 1 h). While small numbers of EdU positive cells were detected in the manubrium (Figs. 1F and 1G), EdU positive cells were uniformly distributed in the umbrella (*Average NND*: uniform, $n = 6/7$, random, $n = 1/7$; Table S1), especially in the exumbrella region (Figs. 1H–1K). By contrast, in the tentacles, large numbers of EdU positive cells were identified as clustered (Figs. 1N and 1O; *NND*: clustered, $n = 7/7$, Table S1). We further confirmed that these EdU-positive cells were proliferating cells using the mitotic marker, anti-Phospho-Histone 3 (PH3) antibody. PH3-positive cells were detected in both the umbrella and the tentacle bulbs (Figs. 1L and 1P). In tentacles, mitotic cells were primarily detected in the ectoderm (Fig. 1P; Fig. S2), whereas in the umbrella, proliferating cells were located in the exumbrella, which was confirmed by the presence of mitotic spindles, detected with an anti-α Tubulin antibody in PH3-positive cells (Fig. 1M).

As observed in young medusae, EdU-positive cells were broadly detected in the entire body of mature medusae (Fig. 2). In the umbrella, EdU-positive cells were more often located in the exumbrella than in the subumbrella, which is similar to the case of young medusae (Figs. 2A–2D). By contrast, in tentacles, EdU-positive cells were restricted to their base, called the tentacle bulb, where two apparent "clusters" are located on both sides of the bulb (Figs. 2E and 2F). These clusters were also observed in the tentacle bulb of young medusae (Figs. 1N and 1O), suggesting that tentacle bulbs may behave as a proliferation zone throughout the medusa stage. Interestingly, in the manubrium of matured medusae, large numbers of EdU-positive cells were detected (Figs. 2G and 2H). This result likely reflects the presence of germ cells that are produced in the manubrium, a feature of the sexually matured *Cladonema* medusa (*Takeda et al., 2018*).

Altogether, our results suggest that cell proliferation may occur uniformly in the medusa umbrella, while a subset of cell proliferation could occur locally in tentacles. Based on these observations, we hypothesized that uniform cell proliferation may control body size growth and tissue homeostasis while clustered cell proliferation in tentacles may contribute to tentacle morphogenesis.

## Cell proliferation is necessary for the control of body size

Animal body size increases upon intake of nutrition because nutrition influences cell proliferation and cell growth (*Bohnsack & Hirschi, 2004*). We first monitored the body size of juvenile medusae by focusing on the size of their umbrella because the umbrella grows in direct proportion with whole body size. Under normal feeding conditions, the medusa umbrella size increased dramatically by 54.8%, from $0.62 \pm 0.02$ to $0.96 \pm 0.02$ mm$^2$ during the first 24 h, with a subsequent minor increase observed over the following 5 days ($0.98 \pm$

0.03 mm$^2$) (Figs. 3A–3C). By contrast, under starved conditions, the size of medusa umbrella did not increase, compared to controls, and rather gradually decreased over the following 5 days (Fig. 3C). Moreover, fewer EdU positive cells were detected in the starved medusae than in fed controls (Figs. 3D–3H; Control: 1,240.6 ± 214.3, Starved: 433.6 ± 133, $t(8) = 3.194$, *$p < 0.05$ ($p = 0.0127$)), suggesting that, at the cellular level, nutrition affects cell proliferation in medusae. These results indicate that body-size growth in juvenile medusae depends on available nutrition.

To test the hypothesis that uniform cell proliferation in medusae contributes to body-size increase, we performed a pharmacological assay to block cell-cycle progression using HU, a cell-cycle inhibitor that causes G1 arrest (*Koç et al., 2004*). Under HU treatment, S phase cells detected by EdU staining disappeared from the medusa body (Figs. 3I–3L). By tracking the size of umbrella, we found that HU-treated medusae did not exhibit the size increase that was observed in controls (Fig. 3M). Together, these results suggest that cell-cycle progression affects body size in *Cladonema* medusae.

## Cell proliferation is necessary for tentacle morphogenesis

In *Clytia hemisphaerica*, another hydrozoan jellyfish, stem-like cells or progenitors are proposed to exist in tentacle bulbs (*Denker et al., 2008*). The clustered or local cell proliferation observed in tentacles, including those in the bulb, of the *Cladonema* medusa may reflect such stem or progenitor cell populations (Figs. 1N–1O and 2E–2F; Fig. S1D). Furthermore, many EdU positive cells were frequently detected in small branched tentacles, which were the most recently branched (Figs. 3D–3E and 3I–3J). To test the hypothesis that local cell proliferation in tentacles contributes to tentacle morphogenesis, we first focused on tentacle branching. Although the initial tentacles have one branch in juvenile medusae, the number of branches gradually increases during medusa maturation (*Fujiki et al., 2019*). In our normal feeding condition, the branching number reached approximately three (2.98 ± 0.05 per tentacle) by day 9 (Figs. 4A and 4C). By contrast, when cell proliferation was blocked with HU, none of the medusae exhibited the typical increase in branched tentacles; rather, all maintained only one branch (Figs. 4B and 4C). Importantly, upon removal of HU, these animals showed an increase in tentacle branching similar to controls, suggesting that the effects of the drug treatment are reversible (Fig. S3C). Combined, these results point to cell proliferation in tentacles as a necessary component for normal tentacle branching.

Cnidarian tentacles have nematocysts, organelles specific to the cnidarian phylum that are utilized for food capture and defense against predators (*Kass-Simon & Scappaticci, 2002*). In *Clytia hemisphaerica*, stem-like cells or progenitors in tentacle bulbs seem to supply nematocysts at the tips of tentacles via cell proliferation, migration to the tip, and differentiation (*Denker et al., 2008*). This evidence raises the possibility that cell proliferation also controls nematocyte development or nematogenesis in hydrozoan jellyfish. To monitor nematocytes in *Cladonema* tentacles, we utilized DAPI, a nuclear staining dye that can label poly-γ-glutamate synthesized in the nematocyst wall (*Szczepanek, Cikala & David, 2002*). Using poly-γ-glutamate staining, we discovered nematocyte size variations ranging from 2 to 110 μm$^2$ (Figs. 4D–4G). We also found that

some of the nematocysts were empty, suggesting that such nematocytes had been depleted (Figs. 4D–4G).

In order to investigate whether cell proliferation in tentacle also contributes to nematocyte maturation, we examined the emptiness of nematocytes after cell-cycle blocking with HU. We detected that the proportion of the empty nematocysts was higher in the medusae with HU treatment than in controls (HU−: 11.4% ± 2.0%; HU+: 19.7% ± 2.0%, Figs. 4D–4G and 4H). This result indicates that even after discharge, nematocytes are still actively supplied by progenitor cell proliferation and that this refill is prevented when cell proliferation is blocked. Taken together, our data suggest that cell proliferation in tentacle plays an important role in both tentacle branching and nematogenesis.

## Cell proliferation is necessary for tentacle regeneration

Cnidarians are known to have a high regenerative capacity (*Galliot & Schmid, 2002*; *Holstein, Hobmayer & Technau, 2003*), and the hydrozoan jellyfish *Cladonema* species exemplifies this typical regenerative ability (*Weber, 1981*). Given the localization of proliferative cells in the tentacle bulb of matured *Cladonema* medusae (Figs. 2E and 2F), we decided to investigate the nature of tentacle regeneration. After dissecting tentacles at their base, we monitored the process of tentacle regeneration (Figs. 5A–5D). During the first 24 h, wound healing occurred at the dissected area (Fig. 5B). Subsequently, the tip of tentacle became elongated and started branching on day 2 (Fig. 5C). At day 4, fully branched tentacles were observed (Fig. 5D), suggesting that tentacle regeneration may follow normal tentacle morphogenesis after elongation.

To examine the initial stage of tentacle regeneration, we examined the distribution of proliferating cells using PH3 staining to visualize mitotic cells. While dividing cells were frequently observed near the amputated area, mitotic cells were dispersed in uncut control tentacle bulbs (Figs. 5E and 5F). We quantified the number of PH3-positive cells present in the tentacle bulbs and found a significant increase in PH3-positive cells in the tentacle bulbs of amputee medusae, compared to controls (Fig. 5G). These observations indicate that initial regenerative responses accompany the active increase of cell proliferation in tentacle bulbs.

In order to test the role of cell proliferation in tentacle regeneration, we blocked cell-cycle progression using HU after dissection and monitored the length of regenerating tentacles. While the tentacles continued to elongate from the bulb structure after dissection in controls, tentacles in animals treated with HU were not able to elongate despite displaying normal wound healing (Fig. 5H). These results demonstrate that cell proliferation in tentacle bulbs is required for proper tentacle regeneration.

## Cell proliferation patterns across different hydrozoan jellyfish

Hydrozoan jellyfish constitute the most broadly varied class of cnidarian jellyfish with approximately 1,150 species worldwide featuring highly diverse morphological and physiological characteristics (*Cartwright & Nawrocki, 2010*; *Schuchert, 2019*). For instance, *Cytaeis uchidae* has four tentacles, and their polyps live exclusively on one type of shell: *Niotha livescens* (*Takeda, Deguchi & Itabashi, 2018*; *Takeda et al., 2013*). Another species,

*R. octopunctata*, has eight grouped-tentacles, and their juvenile medusae asexually produce medusae that grow out of the manubrium (*Berrill, 1952*; *Schuchert, 2007*).

To gain insight into the conserved and diversified nature of cell proliferation in hydrozoan jellyfish, we investigated the spatial pattern of cell proliferation in *Cytaeis* and *Rathkea* medusae. In *Cytaeis* medusa, EdU-positive cells were observed in manubrium, tentacle bulbs, and at the top of the umbrella (Figs. 6A and 6B). PH3-positive cells were also detected in the same regions, suggesting that proliferating cells in *Cytaeis* are distributed in a pattern similar to that observed in *Cladonema*, although there are some discrepancies (Figs. 6C and 6D). By contrast, in *R. octopunctata*, EdU-positive cells and PH3-positive cells were mostly restricted to the manubrium and tentacle bulbs (Figs. 6E–6H). Of note, proliferating cells were frequently detected in the medusa buds that grew out of the manubrium (Figs. 6E–6G), which may reflect asexual reproduction in *Rathkea* medusae. These results suggest that cell proliferation may occur in tentacle bulbs across hydrozoan medusae commonly, while cell proliferation patterns may vary in a species-specific manner with physiology.

## DISCUSSION

In this study, we show that the body size of *Cladonema* medusae is influenced by cell proliferation following uptake of nutrition. Without nutrition and under the blocking of cell-cycle progression, body-size increase is inhibited (Fig. 3). Intriguingly, despite the significant differences between fed and starved animals and between HU-treated and -untreated animals, the body size of *Cladonema* medusae increases during the first 24 h regardless of condition (Fig. 3). A similar body size increase under starved conditions is also reported in the medusa of *Cladonema californicum Hyman* (*Costello, 1998*). These observations can be explained by cell growth via protein synthesis (*Schiaffino et al., 2013*) or accretionary growth, in which cells secrete extracellular matrix to increase extracellular regions, as has been suggested in the growth of cartilage and bone (*Karsenty, Kronenberg & Settembre, 2009*; *Wang, Rigueur & Lyons, 2014*). Given the large amount of collagen that jellyfish contain (*Khong et al., 2016*; *Miura & Kimura, 1985*), extracellular matrix may increase their size during the initial growth of juvenile medusae.

Another interesting feature we observed is that the body size of the starved medusae gradually decreases after 24 h (Fig. 3C). Similarly, upon starvation, *Hydra* polyps cease asexual budding and decrease their size (*Buzgariu, Chera & Galliot, 2008*; *Chera et al., 2009a*), suggesting that cnidarian animals are sensitive to nutrition availability and adapt to metabolic changes. At the organ and tissue level, such size reduction can occur via autophagy or cell death during starvation in diverse phyla (*Jeschke et al., 2000*; *O'Brien et al., 2011*; *Thongrod et al., 2018*; *Tracy & Baehrecke, 2013*). Cnidarians thus may utilize similar mechanisms to reduce cell size and/or cell number to adjust their body size in response to environmental changes. Molecularly, TOR and Hippo signaling are conserved machinery that control organ size, and, as such, these molecules may also play an important role in cnidarian growth control (*Coste et al., 2016*; *Ikmi et al., 2014*; *Loewith & Hall, 2011*; *Van Dam et al., 2011*).

Hydrozoan animals are known to possess interstitial stem cell populations, called i-cells. In *Hydra* and *Hydractinia* polyps, i-cells are localized to the body column, mostly in ectoderm, and have the potential to differentiate into several cell types including nematocytes, nerve cells, and gametes (*Gold & Jacobs, 2013*; *Hemmrich et al., 2012*; *Hobmayer et al., 2012*; *Künzel et al., 2010*; *Müller, Teo & Frank, 2004*). By contrast, the current understanding of the localization and roles of stem-like cells or i-cells in hydrozoan jellyfish are limited (*Leclère et al., 2012*). In *Cladonema* medusa, although mitotic cells are localized in the ectoderm, clusters of proliferative cells are distributed throughout the tentacle, except for the tip region (Figs. S1 and S2). This finding contrasts with the case of *Clytia*, where i-cells are primarily clustered in the tentacle bulb (*Denker et al., 2008*), implying that the i-cells or progenitors of *Cladonema* may be more broadly localized. Our pharmacological experiments confirmed that cell proliferation contributes to tentacle branching, nematogenesis, and tentacle regeneration in *Cladonema* (Figs. 4 and 5), suggesting that these proliferative cells may behave as progenitors or stem-like cells. We further found similar distribution of proliferative cells in tentacle bulbs of *Cytaeis uchidae* and *R. octopunctata* (Fig. 6). Together, these results suggest that the distribution of proliferative cells in tentacle bulbs are widely conserved in hydrozoan jellyfish, while such cells might exist in other tissue to allow body-size increase and species-specific life styles.

## CONCLUSIONS

This study reveals the spatial patterns of cell proliferation during the growth and morphogenesis of the medusae *Cladonema pacificum* at different stages of maturation. Using a cell-cycle inhibitor assay, we show that uniform cell proliferation in the umbrella is responsible for overall body-size increase, while clustered cell proliferation in tentacles contributes to branching morphogenesis, nematocyte differentiation, and regeneration. We further provide evidence for conserved and species-specific cell proliferation patterns in hydrozoan jellyfish by examining two other hydrozoan species, *Cytaeis uchidae* and *R. octopunctata*. The clustered cell proliferation in the tentacle bulbs of these hydrozoan medusae evinces the possibility of a progenitor or stem-like cell population, although its existence will need to be confirmed in future studies. On the whole, our work establishes the basis for understanding the cellular mechanisms of jellyfish growth and homeostasis, which will facilitate future work to dissect the molecular mechanisms underlying these processes.

## ACKNOWLEDGEMENTS

We thank R. Deguchi (Miyagi Univ. Education, Japan) for sharing jellyfish species and helpful discussion. We thank S. Tasaki for helping statistical analyses. We thank H. Takashima for technical assistance. We thank Kuranaga lab members for discussion.

### Funding

This work was supported by the Naito Foundation, the Takeda Science Foundation, the Kanae Foundation for the Promotion of Medical Science, the Daiichi Sankyo Foundation

of Life Science, and the JSPS KAKENHI Grant Numbers JP17H05004 and 17H06332 (to Yu-ichiro Nakajima). The funders had no role in study design, data collection and analysis, decision to publish, or preparation of the manuscript.

### Grant Disclosures
The following grant information was disclosed by the authors:
Naito Foundation, Takeda Science Foundation, Kanae Foundation for the Promotion of Medical Science, Daiichi Sankyo Foundation of Life Science, and JSPS KAKENHI: JP17H05004 and 17H06332.

### Competing Interests
The authors declare that they have no competing interests.

### Author Contributions
- Sosuke Fujita conceived and designed the experiments, performed the experiments, analyzed the data, prepared figures and/or tables, authored or reviewed drafts of the paper, approved the final draft.
- Erina Kuranaga contributed reagents/materials/analysis tools, approved the final draft.
- Yu-ichiro Nakajima conceived and designed the experiments, analyzed the data, contributed reagents/materials/analysis tools, authored or reviewed drafts of the paper, approved the final draft.

### Data Availability
The raw data (pictures) are available in Figshare: Fujita, Sosuke; Nakajima, Yuichiro; Kuranaga, Erina (2019): Raw data for paper (SF-EK-YN). figshare. Dataset. https://doi.org/10.6084/m9.figshare.7935197.v4.

### Supplemental Information
Supplemental information for this article can be found online at http://dx.doi.org/10.7717/peerj.7579#supplemental-information.

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
