# Peer review of "Cell proliferation controls body size growth, tentacle morphogenesis, and regeneration in hydrozoan jellyfish Cladonema pacificum"

_PeerJ, doi:10.7717/peerj.7579_

## Round 0.1 · original submission · Minor Revisions

I have heard back from two reviewers, both of whom were positive about your work, and both of whom recommended minor revisions were needed. I have read over their reviews, and can agree with their constructive comments. Reviewer 1 would like to see more explanations in some sections, while Reviewer 2 wishes for some statistical analyses as well as some more explanations. Thus, combined, the revisions required are extensive. Still, I believe via considering their comments carefully your manuscript can be improved and acceptable for our journal. I look forward to seeing your revised version.

Reviewer 1 ·

Basic reporting

In this manuscript, Fujita et al. analyze the pattern of cell proliferation in the hydrozoan medusa Cladonema pacificum. Using hydroxyurea, they show that cell proliferation is required for medusa growth, tentacle morphogenesis and tentacle regeneration. They also analyze the distribution of proliferating cells in two other hydrozoan medusae. These data add to our understanding of cell proliferation in hydrozoans. The text is clearly written and the figures are relevant and well described.

Correct spelling mistakes: Line 23 ‘cellar’; Line 251 ‘Cnidrians’.

Experimental design

The design of the reported experiments is in general satisfying. However, some details should be added in the Material and Methods and Results parts:

1) In the M&M section, it is written that two types of EdU incubation have been performed, 20uM for 24hr and 150uM for 1hr. It is unclear why the Authors used two different EdU concentrations for the two different incubation times. Also, it is unclear in the Results section which type of EdU labeling has been performed for each experiment. Please add details in the M&M and the Results sections.

2) It is unclear how the medusae were fed. Since the Authors show that feeding has a major impact on medusa growth, it would be important to know the frequency and quantity of feeding in the different experiments. This should be added to the M&M section. Please also explain the meaning of ‘normal feeding condition’ (lines 193, 219).

Validity of the findings

1) In the first section of the Results part, the Authors describe the pattern of cell proliferation in Cladonema pacificum. This description is very succinct and should be much expended. This section should at least answer in details the following questions:
- In which cell layer are the proliferating cells located in the tentacle, manubrium and medusa bell? ectoderm/endoderm? exumbrella/subumbrella?
- Are the proliferating cells uniformly distributed in the tentacles? It seems in Fig 2D that the smallest tentacles contain many more EdU positive cells than the longer ones.
- How are distributed the proliferating cells in the tentacle bulb? Looking at Fig 1B’, it seems that they are mainly located in two “clusters” on either side of the tentacle bulb.
- Are proliferating cells present in the manubrium? And if so where? It is not possible to assess this on the images provided.

2) The Authors use DAPI staining to detect poly-γ-glutamate in Cladonema nematocytes. Line 233-235, the Authors state: “Using poly-γ-glutamate staining, we discovered nematocyte size variations ranging from 2μm2-110μm2 (Fig. 3D). Because nematocytes increase in size during maturation, small nematocysts tend to be immature nematocysts” Implying that the small DAPI labeled nematocytes are the immature forms of the larger ones. This is unlikely since poly-γ-glutamate in both Clytia and Hydra start accumulating only after the nematocyte capsules are fully formed. The authors here are more likely observing different nematocyte types and not different stages of maturation. This could be checked by analyzing the nematocyte composition of the tentacles.

3) The Authors show that Cladonema medusae keep growing for a day after being starved. This has already been described in “Costello J. 1998. Physiological response of the hydromedusa Cladonema californicum Hyman (Anthomedusa: Cladonemidae) to starvation and renewed feeding. J. Exp. Mar. Biol. Eco. 225, 13-28.” Please cite and discuss this article.

Reviewer 2 ·

Basic reporting

See general comments.

Experimental design

See general comments.

Validity of the findings

See general comments.

Additional comments

This paper investigates the role of cell proliferation in growth and regeneration of jellyfish in the hydrozoan cnidarian Cladonema pacifica. The authors used cell proliferation markers EdU and an anti-PhosphoHistone H3 antibody to study the spatial distribution of proliferative cells. In addition, they used hydroxyurea (HU) to pharmacologically inhibit cell proliferation. They find that proliferative cells occur throughout the medusa body, and report a concentration of proliferative cells at the base of tentacles referred to as the tentacle bulb. Moreover, they show that pharmacological perturbation of cell proliferation leads to defects in growth, nematogenesis, and tentacle regeneration in C. pacifica medusae. The authors conclude that C. pacifica medusae possess non-randomly distributed proliferative cells that have necessary roles in development and regeneration.

The role of cell proliferation in jellyfish development and regeneration is understudied, and therefore descriptive and functional analyses of proliferative cells in hydrozoan jellyfish reported in this paper are important. I think this paper should be of interest to broad audiences including invertebrate zoologists, evolutionary developmental biologists and regenerative biologists, and thus would be suitable for publication in PeerJ. Overall, I think that the basic reporting requirements are satisfied for the most part; in particular, figures appear to be of high quality. My main concerns relate to 1) the lack of statistics to support authors’ claim that proliferative cells are uniformly distributed in the umbrella and are enriched in tentacle bulbs, 2) the lack of data to rule out the possibility that phenotypes observed in HU-treated animals have resulted from the toxicity of HU treatment, and 3) the lack of data showing the identity of proliferative cells, as discussed below.

1. On the basis of the spatial distribution of EdU- and/or PH3-positive cells, the authors suggest that proliferative cells are uniformly distributed except in the tentacle bulbs where a cluster of proliferative cells, or “a proliferative zone”, is observed. However, no statistics (e.g. spatial autocorrelation) was performed to reject the null hypothesis that EdU- and/or PH3-positive cells are distributed randomly throughout the medusa body. Evidence is qualitative. The authors should perform appropriate statistical tests to determine whether the spatial distribution of proliferative cells in medusae is uniform, clustered, or random.

2. Based on the phenotype observed in HU-treated animals, the authors conclude that cell proliferation is necessary for normal growth, nematogenesis, tentacle morphogenesis and regeneration in C. pacifica medusae. However, the possibility that the observed phenotype in HU-treated animals resulted from the toxicity of HU treatment has not been ruled out. Are HU-treated animals capable of feeding? Are the effects of HU treatment reversible? Addressing these questions is critical for authors’ main conclusion.

3. As the authors indicate in the manuscript, postembryonic cell types in hydrozoans can be generated by differentiation of interstitial stem cells (i-cells), or by transdifferentiation of differentiated cells (e.g. striated muscle cells in Podocoryne). It would therefore be important to determine the identity of proliferative cells during growth and regeneration in C. pacifica medusae. Do all, or a subset, of the proliferative cells show morphological hallmarks of i-cells (i.e. high nucleo-cytoplasmic ratio, the presence of nucleolus, basophilic, etc)? Where exactly are the proliferative cells located? In the ectoderm and/or endoderm? In the tentacle bulb, are proliferative cells enriched on the oral side, the aboral side, or both?

Minor points:

-Line 240: It is not clear what is meant by “the rate of small nematocysts.” Do authors mean the number or proportion, instead of the rate? Please clarify.
Also, which statistical test was performed to assess the significance of difference between HU-treated animals and the control animals? This information is lacking in Line 264 as well.

-Line 268: Start a new paragraph from “In order to…”

-Line 284: Start a new paragraph from “To gain insight into…”

-Line 310: Start a new paragraph from “Another interesting feature…”

-Figure 2D: How old are the animals?

-Figure 3 caption: E and F are mislabeled. Information on scale bars is missing.

---

## Round 0.2 · accepted · Accept

The manuscript has been well revised, and all comments from reviewers have been addressed. I am happy to move this to production.

Reviewer 1 ·

Basic reporting

no comment

Experimental design

no comment

Validity of the findings

no comment

Additional comments

The Authors did a good job answering the comments from both reviewers. In particular, Figure 1 and the related 'Results' section have been much improved The manuscript can be accepted in its current form, in my opinion.